# Bioinspired Living Coating System in Service: Evaluation of the Wood Protected with Biofinish during One-Year Natural Weathering

**Faksawat Poohphajai** [1,2], **Jakub Sandak** [1,3,*], **Michael Sailer** [4], **Lauri Rautkari** [2], **Tiina Belt** [5] **and Anna Sandak** [1,6]

1. InnoRenew CoE, Livade 6, 6310 Izola, Slovenia; faksawat.poohphajai@innorenew.eu (F.P.); anna.sandak@innorenew.eu (A.S.)
2. Department of Bioproducts and Biosystems, School of Chemical Engineering, Aalto University, P.O. Box 16300, 00076 Aalto, Finland; lauri.rautkari@aalto.fi
3. Andrej Marušič Institute, University of Primorska, Titov trg 4, 6000 Koper, Slovenia
4. Xylotrade B.V, Brinkpoortstraat 28, 7411HS Deventer, The Netherlands; michael@xyhlo.com
5. Production Systems, Natural Resources Institute Finland (Luke), Tietotie 2, 02150 Espoo, Finland; tiina.belt@luke.fi
6. Faculty of Mathematics, Natural Sciences and Information Technologies, University of Primorska, Glagoljaška 8, 6000 Koper, Slovenia
* Correspondence: jakub.sandak@innorenew.eu or jakub.sandak@upr.si; Tel.: +386-40282959

**Abstract:** The service life performance of timber products exposed to natural weathering is a critical factor limiting the broad use of wood as an external building element. The goal of this study was to investigate the in-service characterization of an innovative biofinish coating system. It is a novel surface finishing solution based on the bioinspired concept of living fungal cells designed for effective wood protection. The performance of Scots pine (*Pinus sylvestris* L.) wood coated with biofinish was compared with uncoated references. Samples were exposed to natural weathering for 12 months under the climatic conditions of northern Italy. The visual appearance, colour, gloss, wettability, and 3D surface topography of the wood surface were examined. Results revealed that the total colour changes ($\Delta E$) of biofinish-coated wood were negligible. Untreated Scots pine wood revealed the changes in colour after just three months of exposure. The gloss changes of both surface types were small. The contact angle measured on biofinish-coated wood was higher compared to that of uncoated Scots pine. Surface roughness increased in uncoated wood due to the erosion effect caused by the weathering progress. Conversely, the surface roughness of biofinish-coated samples decreased along the exposure time. This phenomenon was explained by two self-healing mechanisms: migration of non-polymerized oil to the cracked surface, where it polymerizes and creates a closed layer, and local regrowth to cover damaged spots by living fungal cells present in the coating. The obtained results revealed the superior aesthetic performance of the biofinish surface treatment against natural weathering. By considering the fully bio-based nature of the investigated coating, it was concluded that this solution can be an attractive alternative for state-of-the-art wood protection technologies.

**Keywords:** natural weathering; bio-based coating; service life performance; aesthetics; living fungal cells; bioinspired materials design

## 1. Introduction

Weathering is the general term used to describe the slow degradation processes that occur when the material is exposed to the outdoor environment [1–3]. In that case, the deterioration is caused primarily by environmental or abiotic factors and not due to microorganisms. However, the uncoated wood weathering is usually combined with biotic attacks that together may alter the surface appearance. This leads to the creation of green or grey-blue spots on the surface of weathered material which, in association with other

weathering effects, changes the colour of wood towards a grey tonality. Particularly, the weathering processes of wood are triggered by the mutual action of solar radiation and wind-driven rain in combination with several other environmental factors, such as temperature changes, relative humidity (*RH*), wind, air pollutants, molecular oxygen ($O_2$) concentration, or human activities, among others. Surface discolorations, loose fibres, raised grain, checks, cracks, or general roughening are the usual results of wood exposure to natural weathering. Solar radiation is absorbed during weathering by the constitutive woody polymers, including lignin, cellulose, and hemicellulose. Among these components, lignin is recognized as the most sensitive to photodegradation due to its complex structure, revealed to be a variety of functional groups and bonds [1,4]. The photolysis of lignin caused by sunlight leads to the generation of free radicals, which can interact with oxygen to produce hydroperoxides that are converted finally to new lignin chromophores [1,5]. The formation of such chromophores induces the initial yellowing of wood followed by surface colour changes to a silver-grey shade when these chromophore groups are leached out by rain or wind [6,7]. The depth of photodegradation increases with the irradiating light wavelength. Blue light can penetrate wood to a greater extent than violet light. It is capable of bleaching the wood, even if does not alter lignin. Violet light, however, deepens the photodegradation toward wood bulk beyond the zone already affected by the ultraviolet (UV) radiation. Short-wavelength UV radiation causes the photo-yellowing that is directly correlated with the lignin degradation [8]. The detailed mechanisms of natural weathering were the subject of several proceedings studies, including the thorough observation of the alterations at different scales, kinetics, or reactions as well as dose–response models [9,10]. Similarly, diverse approaches were proposed for the characterization of the weathering extent and the determination of the resulting material properties. The colour space coordinates defined by the International Commission on Illumination *CIE L\*a\*b\** and derived $\Delta E$ are frequently used to quantify colour alterations due to lignin degradation and chromophores generation [11–14]. Spectroscopic techniques such as Fourier Transform Infrared Spectroscopy (FTIR) [15,16], Fourier Transform Infrared Spectroscopy–Attenuated Total Reflectance (FTIR-ATR) [17], and/or Fourier Transform Near Infrared Spectroscopy (FT-NIR) [18–20] are often used to investigate changes in functional groups of wood chemical components. Other aspects frequently assessed while evaluating the weathering process of materials include the determination of the surface roughness [21,22], wettability [23,24], or glossiness [18,25,26].

Several surface treatment solutions have been developed to protect wood against weathering-induced deterioration. The application of a coating is one of the most common methods to prevent wood alteration due to its economic- and convenience-based advantages [27]. A coating is defined here as a physical barrier layer that protects wood bulk from environmental factors and microorganisms. The majority of coating solutions can be classified into two categories, including film-forming and various impregnation approaches [2]. Paints and solid-body stains are classic examples of film-forming techniques. Impregnation induces the saturation of the finished subsurface with chemical mixtures containing hydrophobic substances that can penetrate the wood and cures. The most frequently used impregnation products include waxes as well as oil- or resin-based solutions. Although the application of coatings has been widely used for wood protection for a long time, the degradation of the coating layer due to weathering can emerge in a variety of deterioration processes. Some of the most relevant include the photofading of dyed and pigmented polymers [27,28], a loss of gloss and an increase in surface roughness [29], the yellowing of clear coating [30,31], crack formation and peeling [2], or aesthetical problems caused by microorganisms [32–34], among many others. Moreover, an important drawback of modern coating formulations is their composition. Many surface finishing products contain toxic chemicals that might cause negative impacts on the environment and/or can be harmful to living organisms, including humans [35]. Due to the increasing awareness of environmental protection issues, the use of toxic substances allowed until now is increasingly restricted by international legal regulations [36].

Systems and solutions found in nature are a valuable source of inspiration for several advanced applications. Scientists and researchers from different fields use concepts of biomimicking and bioinspiration to solve the most complex challenges and technical tasks. The possibility to benefit from solutions developed by nature recently became interesting for use in cutting-edge materials science and sustainable architecture [37]. Biomimicry aims to understand the fundamental principles of biological processes and adapt these concepts for bio-inspired product applications [38]. New materials and methods for structural design, thermal insulation, and waterproofing can be developed by mimicking the structural, behavioural, functional, and morphological aspects of natural organisms [39]. The largest area of biomimicry research is related to materials, accounting for around 50% of published research [38]. Materials' surface modification includes a wide range of changes related to adjustments of the expected surface energy, appearance, adhesion, and/or inducing unique properties, such as self-regeneration or self-healing. Bioinspired modification can be related here to the generation of a particular surface topography, patterning, activation, or functionalization. Most surfaces are exposed during service life to a diversity of damaging threats, which reduce the surface functionality and require certain actions, such as cleaning, reparation, or replacement. The self-repairing phenomena observed in several biological systems is challenging for replication in traditional building materials since they are simply not alive as biological organisms. However, the development of bio-concrete (where limestone-producing bacteria are activated when a crack occurs) [40] as well as bio-coatings (where living moulds are re-grooving on a wooden surface after damage) provides a pioneering proof of concept for bioinspired solutions demonstrating the novel capacities of building materials [41].

Biofinish technology represents such a bioinspired and environmentally friendly wood treatment solution. The idea of using the functional biofilm produced by a black, yeast-like fungus, *Aureobasidium pullulans*, for the protection of wood, was inspired by the growth of microorganisms on leaves. In industrial applications, the substrate wood was first impregnated with vegetable oil, which serves as a future nutrition source for living *A. pullulans*. Afterwards, the wood's surface was coated with a proprietary biofinish emulsion containing a complete set of ingredients necessary for mould to establish a long-lasting, self-sufficient living biofilm. The proposed solution increases the hydrophobicity of wood substrate due to oil treatment which results in the enhanced dimensional stability of the timber element [41]. The presence of *A. pullulans* living cells plays an important role in protecting wood against other decay from fungi infestation. The colour-giving melanin pigment produced by the fungus provides an aesthetically appealing dark surface on wood, and at the same time, protects the substrate from UV radiation. One of the most attractive aspects of using *A. pullulans* as a living coating system is an induced self-healing ability of such a finished surface. Biofinish is solely composed of natural substances and it is considered a fully sustainable wood treatment solution with minimal environmental impact and low maintenance requirements [42]. Timber products are currently treated at an industrial scale and the do-it-yourself (DIY) coating formulation is available on the market [43]. A combined effect of wood species, oil types, and material climate conditions on the formation of a uniform biofilm were studied previously [42,44,45]. Nevertheless, several aspects regarding protection mechanisms and the durability of biofinish are still poorly understood.

The goal of this study is, therefore, to investigate the in-service performance of biofinish coating during a natural weathering test. A special focus was directed toward the assessment of the alteration of appearance and aesthetical properties, to be evaluated with an objective multi-sensor approach and confronted with a reference timber product frequently used in a similar configuration.

## 2. Materials and Methods

### 2.1. Experimental Samples and Natural Weathering

Thirty wooden blocks made of Scots pine (*Pinus sylvestris* L.) wood with dimensions of 150 mm × 75 mm × 20 mm (length × width × thickness, respectively) were used as experimental samples. Half of the sample set (15 blocks) was coated with biofilm (Biofinish, Xyhlo B.V. Deventer, The Netherlands) following the industrial procedure of the coating application (0.125 L·m$^{-2}$). Two layers of coating were applied penetrating wood subsurface to the depth of approximately 350 μm. The remaining set of fifteen Scots pine blocks was left uncoated and served as a reference. Thirty samples were analysed in total, including two finishing types, five natural weathering scenarios, and three replicas for each scenario.

Natural weathering tests were performed in San Michele, Italy (46°11′15′′ N, 11°08′00′′ E), for a total period of 12 months, starting in March 2017. The objective of this test was to collect a set of reference data regarding material performance in a function of the exposure time. Samples were exposed on a vertical stand corresponding to a typical building façade configuration. The stand was oriented to face the southern direction. The representative meteorological data for the studied location are summarized in Table 1. Three replicas were collected from the stand each third month to terminate the deterioration progress. Thus, a collection of samples exposed for 0, 3, 6, 9, and 12 months was gathered as a result of the experimental campaign. Each exposure period corresponds to a single experimental scenario. Three replica samples allowed insight into the reliability and repeatability of results. All samples were stored after collection from the stand in a climatic chamber (20 °C, 65% *RH*) to stabilize conditions before follow-up measurements.

**Table 1.** Weather data (average values) acquired during weathering progress.

| Exposure Month | 1 | 2 | 3 | 4 | 5 | 6 | 7 | 8 | 9 | 10 | 11 | 12 |
|---|---|---|---|---|---|---|---|---|---|---|---|---|
| Mean Temp (°C) | 12 | 14 | 19 | 24 | 25 | 25 | 17 | 14 | 7 | 1 | 4 | 4 |
| Days of Frost | 0 | 0 | 0 | 0 | 0 | 0 | 0 | 8 | 5 | 24 | 16 | 10 |
| Total Rain (mm) | 32 | 108 | 50 | 36 | 27 | 17 | 6 | 1 | 28 | 3 | 1 | 7 |
| *RH* (%) | 49 | 45 | 51 | 51 | 50 | 52 | 68 | 62 | 69 | 74 | 82 | 55 |
| Mean Wind Speed (km/h) | 7 | 9 | 7 | 9 | 8 | 6 | 5 | 5 | 4 | 3 | 3 | 5 |

### 2.2. Colour Measurement

Experimental samples were scanned with the office scanner HP Scanjet G2710 (Palo Alto, CA, USA). The colour changes were determined using portable MicroFlash 200D spectrophotometer (DataColor Int, Lawrenceville, NJ, USA). The selected illuminant was D65, and the viewer angle was 10°. The colour was measured at ten given spots that were randomly selected over the surface of investigated wood specimens. Colour changes were evaluated following the *CIE L\* a\* b\** colour space system where colour is expressed by three parameters: *L\** (lightness), *a\** (red-green tone), and *b\** (yellow-blue tone). The total colour change Δ*E* was calculated according to Equation (1):

$$\Delta E = \sqrt{(\Delta L^*)^2 + (\Delta a^*)^2 + (\Delta b^*)^2}, \tag{1}$$

where Δ*L*, Δ*a*, Δ*b* correspond to differences between colour coordinate values measured at the given time and referenced to the corresponding value of initial colour.

### 2.3. Gloss Measurement

The mode of light reflection from the evaluated surfaces was measured using a REFO 60 (Dr. Lange, Düsseldorf, Germany) portable gloss meter with incidence and reflectance angles of 60°. Ten measurements were performed on each wood sample in parallel and perpendicular to the fibre directions. An average value was considered as a measure of the gloss, with both minimum and maximum values, determining the observation range and repeatability of the assessment.

### 2.4. Wettability

Dynamic contact angle measurement when wetting the surface with distilled water was performed with optical tensiometer Attension Theta Flex Auto 4 (Biolin Scientific, Gothenburg, Sweden). Five measurements were performed on each specimen, implementing the sessile drop measurement approach. The nominal volume of each drop was 4 μL. The volume was precisely controlled by both droplet dispenser and a digital image analysis tool integrated with instrument software. The measurement of the drop contour was initiated at the moment of the drop contact with the assessed sample surface. The droplet image acquisition lasted for 20 s. Images were post-processed with the proprietary software (OneAttension v.4.0.5) of the tensiometer. The contact angle was determined by implementing the Laplace equation. The contact angle observed at the third second of the test was assumed as a representative quantifier. Five replica measurement results were averaged to reduce the scatter of results. The range of observed values (minimum–maximum) was used to define the variability and reliability of the contact angle assessment.

### 2.5. Microscopic Observation and 3D Roughness Measurement

Keyence VHX-6000 digital microscope (Keyence, Osaka, Japan) was used for microscopic observations and 3D surface topography assessment. Colour images were acquired with magnifications of ×30 and ×1000. The light configuration and its intensity were adjusted to assure an optimal image quality and elimination of the saturated pixels. High magnification images were acquired in 3D depth reconstruction mode. It allowed post-processing of data to extract representative surface profiles as well as computation of the surface roughness indicators. An area of 20 mm × 20 mm was assessed by implementing a stitching algorithm for acquired 3D images. The proprietary software (VHX-6000 v.3.2.0.121) of the Keyence microscope was used for the surface topography post-processing. The error of form was initially removed by extracting an average plane. The resulting 3D topography map was filtered with a Gaussian band pass filter (2 μm < λ < 2.5 mm). Arithmetical mean height ($S_a$), skewness ($S_{sk}$), and kurtosis ($S_{kt}$) were computed as the surface irregularity quantifiers.

## 3. Results and Discussions

### 3.1. Surface Visual Appeal

Appearance images of the investigated samples are presented in Figures 1 and 2 for reference (uncoated) and biofinish-coated samples, respectively. Natural Scots pine wood becomes darker after the initial three months of exposure, followed by a slight lightening and converting to the grey tonality. The grey colour becomes dominant after weathering for one year. Small surface cracks become visible to the naked eye from month 9. At the same time, big cracks were present together with raised fibres and eroded surface marks. The appearance of samples coated with biofinish does not change, including consistent impressions of the colour, roughness and overall surface integrity.

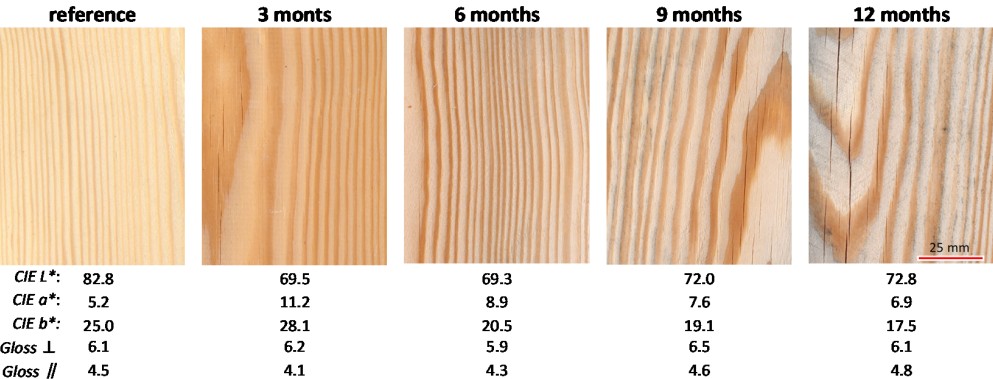

| | reference | 3 monts | 6 months | 9 months | 12 months |
|---|---|---|---|---|---|
| *CIE L\**: | 82.8 | 69.5 | 69.3 | 72.0 | 72.8 |
| *CIE a\**: | 5.2 | 11.2 | 8.9 | 7.6 | 6.9 |
| *CIE b\**: | 25.0 | 28.1 | 20.5 | 19.1 | 17.5 |
| *Gloss* ⊥ | 6.1 | 6.2 | 5.9 | 6.5 | 6.1 |
| *Gloss* ∥ | 4.5 | 4.1 | 4.3 | 4.6 | 4.8 |

**Figure 1.** Surface texture, *CIE L\*a\*b\** colour coordinates and glossiness parameters altered during weathering progress of uncoated Scots pine samples.

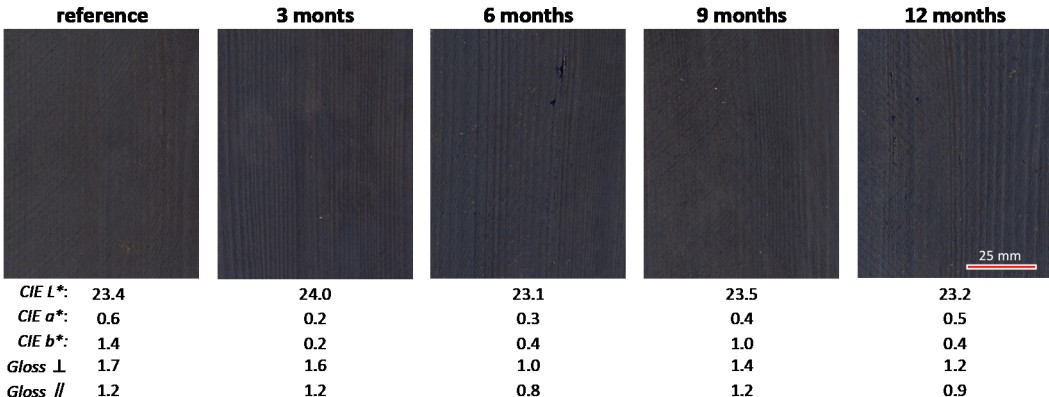

| | reference | 3 monts | 6 months | 9 months | 12 months |
|---|---|---|---|---|---|
| *CIE L\**: | 23.4 | 24.0 | 23.1 | 23.5 | 23.2 |
| *CIE a\**: | 0.6 | 0.2 | 0.3 | 0.4 | 0.5 |
| *CIE b\**: | 1.4 | 0.2 | 0.4 | 1.0 | 0.4 |
| *Gloss ⊥* | 1.7 | 1.6 | 1.0 | 1.4 | 1.2 |
| *Gloss ∥* | 1.2 | 1.2 | 0.8 | 1.2 | 0.9 |

**Figure 2.** Surface texture, *CIE L\*a\*b\** colour coordinates and glossiness parameters altered during weathering progress of biofinish-coated Scots pine samples.

### 3.2. Surface Colour

The progress of *CIE L\*a\*b\** changes over the duration of the natural weathering test is shown in Figure 3. The average *CIE L\** of the uncoated Scots pine samples decreased during the first three months of exposure, to become constant afterwards (Figure 3a). The average *CIE a\** of the uncoated wood increased at the initial stage of exposure, but after that, it steadily decreased until the end of the natural weathering process (Figure 3b). A similar trend was noticed in the case of the *CIE b\** coordinate (Figure 3c). Both, *CIE a\** and *b\** asymptotically followed the horizontal axis, confirming the overall tendency for greying the weathered surface outlook. A very different trend of the *CIE L\*a\*b\** colour changes was noticed for coated samples. All investigated colour indicators were constant along the whole test duration. Correspondingly, the scatter of the *CIE L\*a\*b\** values within all replicas was substantially smaller than the same scatter observed in the uncoated wood samples. It was confirmed by the total colour changes (ΔE) analysis present in Figure 3d. The ΔE of uncoated Scots pine rapidly increased during the initial three months of exposure. The same quantifier steadily decreased afterwards until the end of the exposure period of twelve months. Biofinish-coated specimens revealed much lower values of ΔE compared to those noticed in uncoated Scots pine. ΔE was relatively constant with minor variations. The nature of ΔE changes in the coated wood samples suggests the source of the analysed discoloration to be related to the intrinsic variance of the colour within tested samples rather than the weathering-induced alterations.

The decrease in *CIE L\** of uncoated Scots pine at the initial stage of exposure indicates that the darkening of the wood surface is due to the accumulation of degradation products of lignin [2] followed by the infestation of the wood surface by fungi [2,27]. The initial increase in *CIE b\** is also associated with the degradation of lignin [46]. Consequently, the follow-up decreases in yellowness (*CIE b\**) may be attributed to the leaching of decomposed lignin and its extractives by water [47]. The changes in *CIE a\** values are primarily determined by the changes of the chromophore groups present in some wood extractive components [48]. The formation of additional chromophoric systems was found to be triggered by the solar radiation interaction with the wood chemical components [2,49]. The overall mechanism of wood discolouration induced by the weathering processes follows three overlapping phases. An initial darkening, yellowing or browning of wood is observed as a result of the quinone formation and accumulation of degradation products of lignin. The greyish tendency of weathered wood after a long-term exposure period is associated with the leaching out of extractives and cleavage products of lignin photodegradation. Finally, the dark-grey colour of the long-term weathered wood surface is a result of surface infestation by fungal hyphae, spores and derived pigments [49,50].

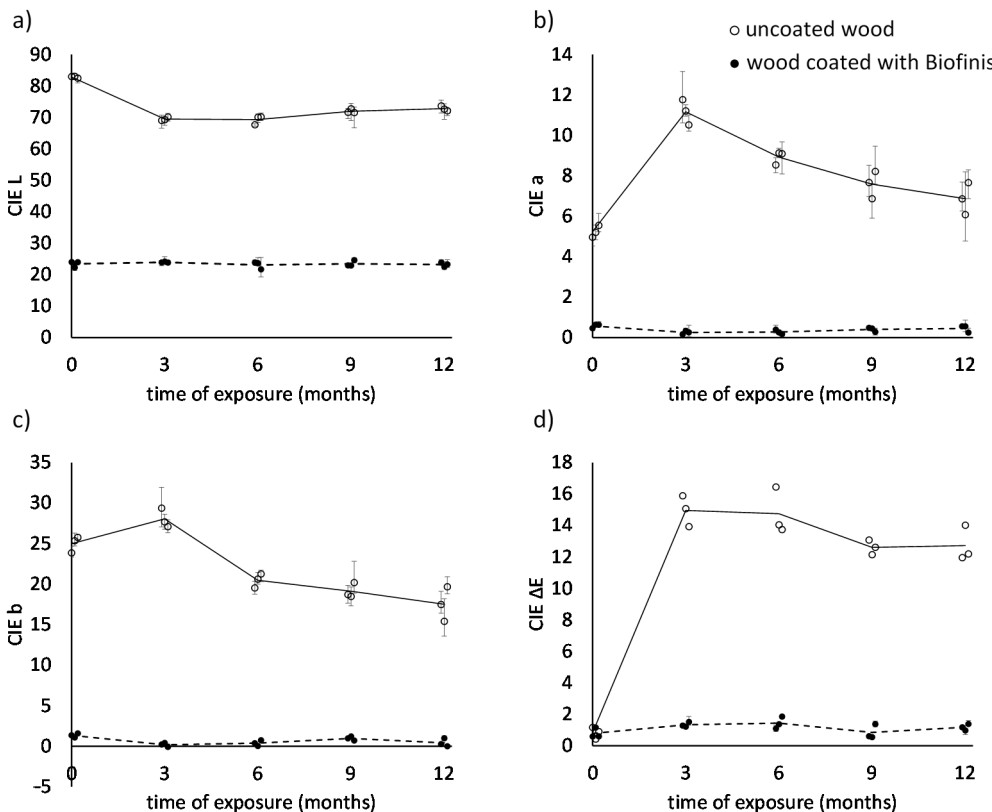

**Figure 3.** *CIE L\*a\*b\* colour coordinates of uncoated Scots pine and wood coated with biofinish as a function of exposure time. Note: Error bars correspond to the range (minimum–maximum) of results observed:* (**a**) *CIE L\*,* (**b**) *CIE a\*,* (**c**) *CIE b\*,* (**d**) *CIE ΔE.*

Presence of moulds depends on the intrinsic properties of the specific wood substrate. Heartwood is considered to be less susceptible to decay fungi than sapwood. In the case of Scots pine heartwood, it is due to the presence of extractive components possessing fungicide capacity, which can suppress fungal growth [51]. The relative humidity and temperature of the ambient air are two factors regulating the moisture content of wood. Wood moisture is known as a critical factor determining the ability of microorganisms for initial surface colonization and further growth. The threshold limit for mould growth is usually at the level of wood moisture content of 20% and corresponding air relative humidity of 75–80%. However, the actual fungal infestation risk highly depends on the exposure time to the elevated moisture, ambient air temperature, inherent susceptibility of the exposed surface to moulds, and presence of specific mould species in the ambient air [52].

Negligible variations of the *CIE L\*a\*b\** colour coordinates of wood coated with biofinish can be explained by the presence of colour-giving melanin pigment produced by *A. pullulans*. Melanin, in addition to colour determination, functions as a barrier layer protecting the wood surface against alien microorganisms. Melanin production by the fungi does not affect their growth or development but enhances survival under environmental stresses. These include protection against UV irradiation, the accumulation of pollutants (such as heavy metals), tolerance to extreme temperatures, desiccation or high salts concentrations, as well as resistance to the cell wall-degrading enzymes and other pathogens [53–56]. Several studies confirm the stimuli effect of UV radiation on the enhanced melanin production by the *A. pullulans* [57–59].

*3.3. Gloss Evaluation*

The gloss measured parallel and perpendicular to fibres on uncoated and biofinish-coated Scots pine specimens was constant along the natural weathering process (Figure 4).

The average gloss value measured in parallel to fibres of uncoated Scots pine oscillated around 6, while those measured perpendicularly was approximately 5. Gloss values were smaller for samples coated with biofinish and corresponded to 1 in both measurement directions. The spread of data defined as the difference between the minimum and maximum recorded values was higher for gloss measured parallel to the fibres in the case of both surface finish configurations.

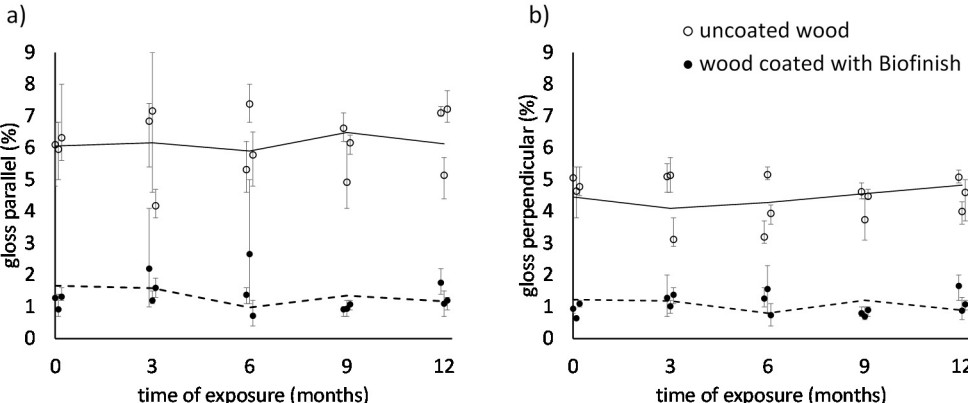

**Figure 4.** Gloss changes of uncoated Scots pine and wood coated with biofinish as a function of exposure time. Note: Error bars correspond to the range (minimum–maximum) of the results observed: (**a**) gloss parallel, (**b**) gloss perpendicular.

Several studies are reporting a decrease in wood glossiness associated with exposure to natural weathering [18,20,60,61]. It was not directly confirmed by the present experimental results that are related to the relatively short period of the test as well as the initial surface configuration that was intended to be moderately rough. The reduction in the wood surface glossiness is associated with the abrasion of wood surfaces and the accompanying surface erosion process. The wood surface usually becomes rougher as a result of the weathering which enhances the diffuse (Lambertian) light reflectance component. Consequently, the scatter of light on the surface become higher, which is recorded as a glossiness value reduction [49].

*3.4. Wettability*

The change of average wettability values expressed as a contact angle θ induced by the natural weathering process are shown in Figure 5. The contact angle of biofinish-coated wood was twice as high (~120°) as that measured on uncoated Scots pine surface (~60°). The contact angle of both types of specimens decreased during the initial three months of exposure. However, the drop was very apparent in the case of uncoated wood, resulting in θ < 20°, which was stable until the end of the weathering test. It is associated with chemical changes of the weathered wood surface at the initial period of exposure, as well as the leaching of extractives possessing hydrophobic nature. Such a small θ corresponds to the instant surface wetting and spill out of the water droplet after contact with the measured surface. The contact angle measured on the surface of biofinish-coated wood was steadily decreasing along the natural weathering test duration. This leads to the conclusion that wood hydrophobicity decreased along the exposure time, even if the surface coating substantially reduced the kinetic of such changes. Similar observations were reported by other researchers for Scots pine [62] and other wood species [62–64]. The increased wettability of the weathered wood surface even further accelerates the subsurface deterioration. This eases the leaching of extractives and other degradation products of lignin. Elevated surface moisture content triggers the loosening of the cellulose fibres that trigger micro-cracks presence and an increase in the roughness [65].

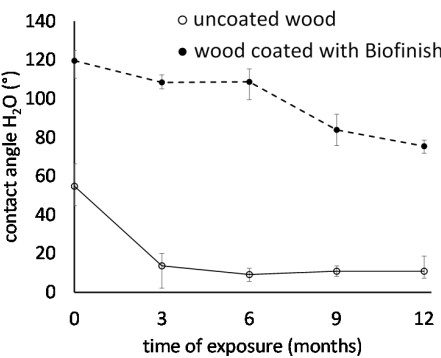

**Figure 5.** Changes in the contact angle of uncoated Scots pine and biofinish-coated wood as a function of exposure time. Note: Error bars correspond to the range (minimum–maximum) of results observed.

### 3.5. Surface Topography

Changes in surface topography for uncoated and biofinish-coated samples are presented in Figures 6 and 7, respectively. An increase in the surface roughness was mostly observed in uncoated pine wood specimens, where all surface topography descriptors ($S_a$, $S_{sk}$ and $S_{kt}$) changed during weathering progress (Figure 6). The overall curvature changes of the surface topography profiles were related to the wood deformations induced by the moisture content variations and consequently to wood shrinkage/swelling. The biofilm coating layer affected the thermodynamic properties of the wood subsurface limiting the absorption and diffusion of moisture. An increase in the roughness was related to the natural weathering processes and resultant photolysis and/or hydrolysis of the constitutive wood polymers, particularly lignin. As a result, single fibres were removed from the wood matrix and degradation components are leached out after rain events. This leads to the wood surface erosion revealed as an increase in the surface roughness [66].

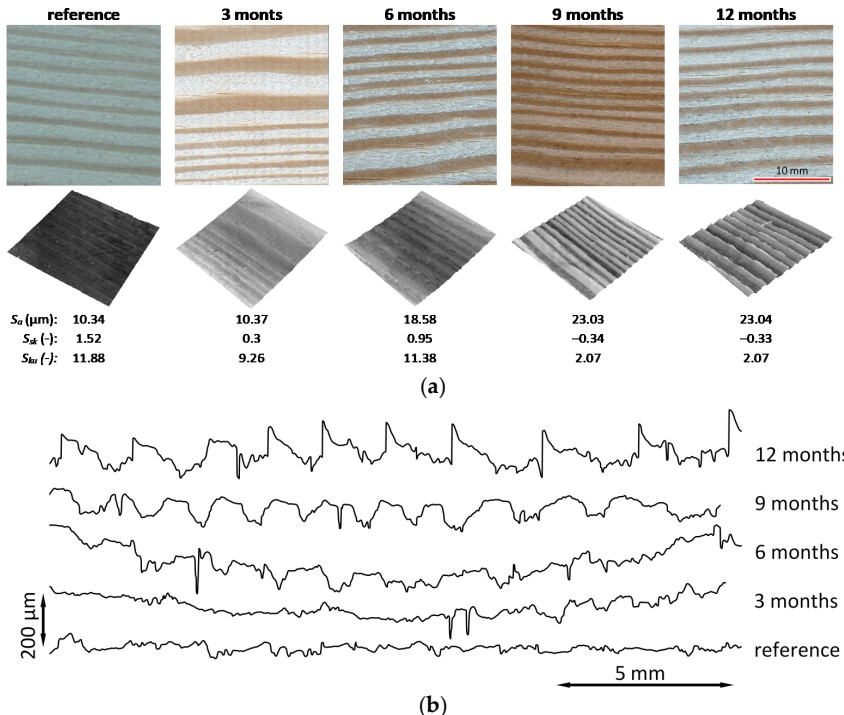

**Figure 6.** 3D surface topography map, typical surface profiles and high magnification images of the Scots pine wood exposed to natural weathering for 12 months, (**a**) high magnification images and 3D surface topography map, (**b**) typical surface profiles.

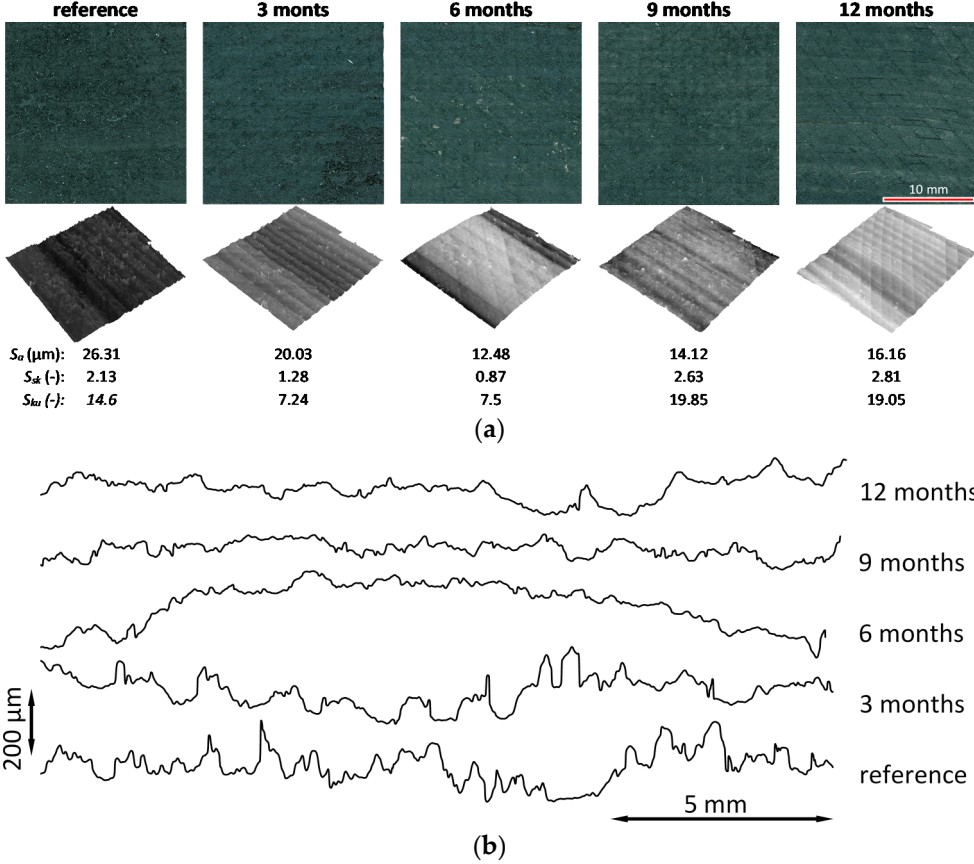

**Figure 7.** 3D surface topography map, typical surface profiles and high magnification images of the Scots pine wood coated with biofinish exposed to natural weathering for 12 months, (**a**) high magnification images and 3D surface topography map, (**b**) typical surface profiles.

Opposite trends of surface roughness alteration were observed for wooden samples coated with biofinish. The arithmetical mean surface height ($S_a$) decreased with the weathering progress, indicating that the surface became smoother than at the initial stage (Figure 7). Such flattening of the wood surface can be explained by the gradual secretion of linseed oil that was used for sample impregnation before coating application. It should be mentioned that the preferred surface for the application of the biofinish product is rough. The presence of surface irregularities eases adhesion to the coating emulsion as well as fungi spores. Furthermore, living cells of *A. pullulans* are capable of regenerating and regrowing on the wooden surface after the initial application of the coating. This leads to the apparent roughness reduction by filling surface micro-voids and other irregularities caused by the fungal maturation and development. This is confirmed by the visual analysis of surface topography profiles. While an evident roughening of the surface outline is noticeable for uncoated wood, the excessive initial waviness disappears on the biofinish-coated profiles.

Low values of skewness, $S_{sk} \approx 0$, as observed for the uncoated pine samples, indicates a normal distribution of the irregularity deviation. However, a shift in the skewness toward the higher side of the topography map is typical of porous materials, such as softwoods. The decrease in skewness observed for the uncoated wood can be associated with the loss of fibres and the general progress of the weathered wood surface erosion. Kurtosis ($S_{kt}$) is a measure of the topography histogram profile sharpness. Values of $S_{kt}$ of uncoated pine wood gradually decreased from platykurtic ($S_{kt} \approx 12$) to leptokurtic ($S_{kt} \approx 2$). The kurtosis of the coated wood surface was continuously elevated, indicating its strong platykurtic nature. The coated surface possessed a balanced distribution of peaks (loose fibres) and valleys (surface cracks).

### 3.6. Microscopic Observations

Microscopic images of investigated samples are shown in Figure 8. The development of mould fungi with dark-coloured hyphae and spores is a very common phenomenon observed on coated and uncoated wooden façades. Moulds and blue stain fungi are considered to be undesirable elements, especially on light-coloured wooden façades [21]. Consequently, the presence of such microorganisms may substantially reduce the aesthetical service life of the whole building. Moreover, the wood decay process can be initiated by the presence of fungal spores or mycelia fragments on wood substrate. Spores, as indicated by arrows in Figure 8, will germinate under favourable conditions to produce fine hair-like structures known as fungal hyphae. Hyphal fragments landing on the wood surface can in many cases initiate fungal growth, leading to a broader colonization. *Ascomycetes* fungi, including the *A. pullulans* studied in this research, belong to soft rot species. These present a mixture of the white and brown rot characteristics digesting both carbohydrates (cellulose and hemicellulose) as well as lignin. It was found that the structural damage caused by the *A. pullulans* fungus is constrained to the external few millimetres of the exposed wooden element outline [67]. *A. pullulans* possesses nutritional versatility, which is associated with its wide enzymatic profile [68]. Therefore, the potential damage of a wooden surface can be eliminated by providing a preferable nutrition source that is different from the substrate wood. Linseed oil was used in the case study. It should also be mentioned that *A. pullulans* is known to possess antagonistic properties towards other yeasts and fungi. The biofilm developed by this fungus may inhibit the growth of non-desired microorganisms, assuring supplementary protection against biotic factors [69].

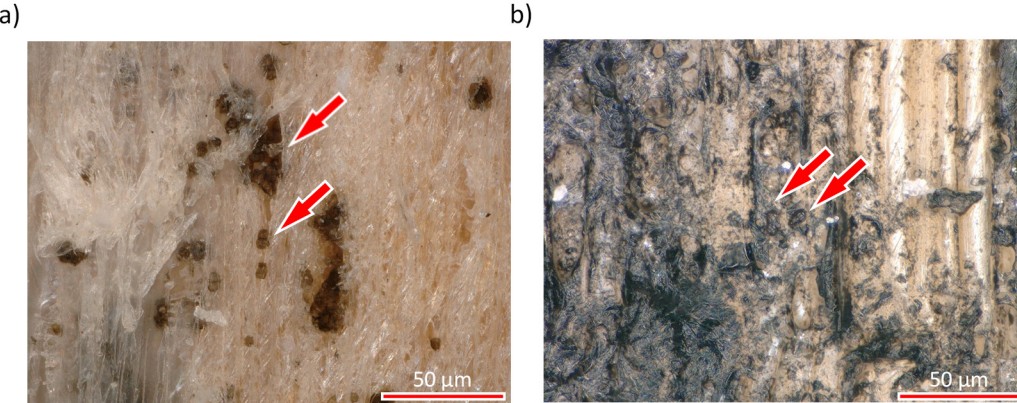

**Figure 8.** Chlamydospores of *A. pullulans* on the surface of uncoated Scots pine after 9 months of exposure (**a**) and a biofinish-coated reference sample not exposed to natural weathering (**b**), observed under microscope with ×1000 magnification.

## 4. Conclusions

A majority of the common surface treatments, coating products, or finishing techniques rely on mineral oil binders and diverse ingredients that may not be considered environmentally friendly. This study focused on the assessment of the in-service performance of the innovative coating system based on a bioinspired concept. The performance of Scots pine (*Pinus sylvestris* L.) wood coated with biofinish was confronted with uncoated reference wood following a 12-month natural weathering trial. Experimental samples were objectively characterized following a multi-sensor approach, with a special focus on aesthetical deterioration aspects. Performance of the wood treated with biofinish was superior when compared to the uncoated reference regarding all examined aspects. An interesting phenomenon related to the dual self-healing mechanism was observed as the surface texture consistency and proved by an evidenced decrease in the surface roughness along with the weathering progress. Such a trend is contrary to the majority of materials exposed to weathering that are gradually deteriorating and eroding. An entirely bio-based composition of the biofinish coating solution is a great advantage assuring unique sustain-

ability and compatibility to the natural environments. It can be considered, therefore, as an attractive alternative for state-of-the-art wood surface protection solutions.

## 5. Patents

Ecologically protected material EP1704028 B1 & US7951363 B2.
PCT patent application 2020: PCT/NL2020/050585.

**Author Contributions:** Conceptualization, F.P., J.S., M.S., L.R., T.B., and A.S.; methodology, J.S. and A.S.; software, J.S.; validation, F.P., J.S., and A.S.; formal analysis, F.P., J.S., and A.S.; investigation, F.P., J.S., M.S., L.R., T.B., and A.S.; resources, J.S. and A.S.; data curation, J.S. and A.S.; writing—original draft preparation, F.P., J.S., and A.S.; writing—review and editing, F.P., J.S., M.S., L.R., T.B., and A.S.; visualization, J.S.; supervision, J.S., L.R., T.B., and A.S.; project administration, A.S.; funding acquisition, J.S. and A.S. All authors have read and agreed to the published version of the manuscript.

**Funding:** The authors gratefully acknowledge the European Commission for funding the InnoRenew project (Grant Agreement #739574 under the Horizon2020 Widespread-2-Teaming program), the Republic of Slovenia (investment funding from the Republic of Slovenia and the European Regional Development Fund) and infrastructural ARRS program IO-0035. Part of this work was conducted during project BIO4ever (RBSI14Y7Y4), funded within call SIR by MIUR-Italy; the project Multi-spec (BI-IT/18-20-007), funded by ARRS-Slovenia; Archi-BIO (BI/US-20-054) funded by ARRS-Slovenia, J7-9404 (C) funded by ARRS-Slovenia, and CLICK DESIGN, "Delivering fingertip knowledge to enable service life performance specification of wood", (No. 773324) supported under the umbrella of ERA-NET Cofund ForestValue by the Ministry of Education, Science and Sport of the Republic of Slovenia. ForestValue has received funding from the European Union's Horizon 2020 research and innovation programme.

**Institutional Review Board Statement:** Not applicable.

**Informed Consent Statement:** Not applicable.

**Data Availability Statement:** The data presented in this study are available upon request from the corresponding author.

**Acknowledgments:** The experimental samples were provided by Xyhlo B.V., the Netherlands. The support of the industrial partner is highly appreciated.

**Conflicts of Interest:** The authors declare no conflict of interest.

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
