# Peer review of "Bioinspired Living Coating System in Service: Evaluation of the Wood Protected with Biofinish during One-Year Natural Weathering"

_coatings, doi:10.3390/coatings11060701_

Round 1

Reviewer 1 Report

In this paper, authors evaluated the performance of bioinspired coating system. The properties, including visual appearance, colour, gloss, wettability, and 3D surface topography, of the wood surfaces with and without the innovative biofinish coating are studied. Based on the results, the biofinish surface treatment revealed superior aesthetic performance against natural weathering for 12 months.  This method provides a potential solution to protect wood with bioinspired coating system. I propose the acceptance of this article only after the following questions are addressed.

Comments:

  1. Figure 2 the resolutions of subfigures are not good especially for the picture of the surface texture after 12 months.
  2. What’s the thickness of the biofinish coating on the wood? Please add the value in the method section. Does the thickness of the coating make an influence on the final evaluation? Please clarify.
  3. Add the scale bar on Figure1, Figure2, Figure6, and Figure7.
  4.  From Figure8.b, the coated biofinish film was abrased after 9 months so that the wood exposure outside.  Please clarify if it affected the colour coordinates of wood coated with biofinish along the time.
  5. In Figure5, the contact angle of wood without coating didn’t change much after 3 months of exposure. Please clarify the potential reason.
  6. For the surface topography, after 6 months, the surface of wood looks like a concave plane, while the surface of the wood with coating likes like a convex plane. Please clarify the potential reason.

Author Response

Reviewer 1

In this paper, authors evaluated the performance of bioinspired coating system. The properties, including visual appearance, colour, gloss, wettability, and 3D surface topography, of the wood surfaces with and without the innovative biofinish coating are studied. Based on the results, the biofinish surface treatment revealed superior aesthetic performance against natural weathering for 12 months.  This method provides a potential solution to protect wood with bioinspired coating system. I propose the acceptance of this article only after the following questions are addressed.

Thank you very much for your kind evaluation of our manuscript and your time dedicated to the review. We have modified the manuscript according to yours (and other Reviewer) suggestions. Please find attached the revised version of our work with track changes. There are also some responses to your letter as described below:

Comments:

1. Figure 2 the resolutions of subfigures are not good especially for the picture of the surface texture after 12 months.

A separate file with the source of figures in the vector (native format) are provided in addition to the metafiles used in the current version of the manuscript.

2. What’s the thickness of the biofinish coating on the wood? Please add the value in the method section. Does the thickness of the coating make an influence on the final evaluation? Please clarify.

Investigated samples were industrially coated with two layers of biofinish product. The coating protocol included also surface impregnation by the oil. It is very difficult to determine the thickness of biofilm as it varies depending on the specific viability of the microorganisms as well due to nature of the coated substrate (wood). We determined the thickness of the coating layer (including oil impregnated substrate by measuring the colour gradient (difference between coated (coloured) and uncoated (natural wood colour) surface on the exposed section of the sample subsurface. The estimated coating penetration/impregnation depth was ~350 µm. An additional sentence was included in the revised manuscript text.

3. Add the scale bar on Figure1, Figure2, Figure6, and Figure7.

Thank you, it is an important suggestion. The scale bar is included in all figures containing images of experimental samples.

4.  From Figure8.b, the coated biofinish film was abrased after 9 months so that the wood exposure outside.  Please clarify if it affected the colour coordinates of wood coated with biofinish along the time.

Perhaps the caption of the Figure was not clear, as Figure 8b presents microscopic image of the reference sample (not exposed to weathering). For that reason, the biofilm is not yet fully developed. Moreover, we observed that along the progress of weathering and in particularly exposure of investigated coating system to UV radiation stimulated growth of A. pullulans and production of melamine pigment.

 5. In Figure5, the contact angle of wood without coating didn’t change much after 3 months of exposure. Please clarify the potential reason.

The additional sentence clarifying the reason was added to the manuscript text: “It is associated with chemical changes of the weathered wood surface at the initial period of exposure, as well as leaching of extractives possessing hydrophobic nature.” It should be stated that the drop was nearly fully absorbed after the initial 3 seconds following initial wetting that resulted in a very low contact angle.

6. For the surface topography, after 6 months, the surface of wood looks like a concave plane, while the surface of the wood with coating likes like a convex plane. Please clarify the potential reason.

The weathered samples’ surface deformations are related to the moisture content gradient and related shrinkage/swelling of wood. The biocoating affected the diffusion and absorption coefficients, therefore the moisture content dynamics was different than in uncoated wood. We included following explanation, as requested by reviewer:

“The overall curvature changes of the surface topography profiles were related to the wood deformations induced by the moisture content variations and consequently to wood shrinkage/swelling. The biofilm coating layer affected the thermodynamic properties of the wood subsurface limiting absorption and diffusion of moisture.”

We hope that the manuscript has been improved after considering your and other Reviewer comments. We do hope that it may be re-considered for publication in the Coatings journal.

With sincerely regards

Jakub Sandak on behalf of authors

Reviewer 2 Report

Revision of the MS - Coatings- “Bioinspired living coating system in service: Evaluation of the wood protected with biofinish during one-year natural weathering

The MS presented by the authors is an interesting paper on a innovative coating for wood. The results are interesting and well detailed. Overall, the different sections of the paper are well distributed and I express a general positive evaluation. Some corrections and simplifications are necessary.

1. Line 41-42. Among the factors to be considered in deterioration, it should be specified how this is influenced by the interaction of the factors mentioned and the wood essence. Moreover, in the sentence you have to consider the biotic effect on deterioration.

2. Line 50-51. Insert a bibliographic reference that highlights the differences you have explained.

3. Line 54-55. The sentence is a bit general as sometimes with the ultra violet component of sunlight you can have a loss of yellowing (for example in teak wood see Drevo12004 (woodresearch.sk))

4. Line 60. Minimize the use of abbreviations only and add the full name for the abbreviations at least for the first time. CIELAB color space also referred to as L * a * b *

5. Line 113. Aureobasidium pullulans and A. melanogenum are different species or the same species ?. Specify if there are multiple species used as biofilms

6. Line 139-146. Specify if the starting material have the same origin and cutting age, and how the blocks have been handled (drying, cutting, etc.).

7. Line 152. Specify how climate data was collected. If dataloggers are used what were the measurement intervals.

8. Line 172-173. Provide more information on the equipment used. Is it a portable or benchtop instrument?;

9. Line180. Attension ®  not attention;

10. Line 192. Specify if tangential sections of reduced thickness have been obtained for observation under the microscope;

11. Line 216-219. The figure 1 and 2 can be grouped into a single figure;

12. Line 390. In fig. 8 (b) is missing;

Author Response

Reviewer 2

The MS presented by the authors is an interesting paper on a innovative coating for wood. The results are interesting and well detailed. Overall, the different sections of the paper are well distributed and I express a general positive evaluation. Some corrections and simplifications are necessary.

Thank you very much for your kind evaluation of our manuscript and your time dedicated to the review. We have modified the manuscript according to yours (and other Reviewer) suggestions. Please find attached the revised version of our work with track changes. There are also some responses to your letter as described below:

1. Line 41-42. Among the factors to be considered in deterioration, it should be specified how this is influenced by the interaction of the factors mentioned and the wood essence. Moreover, in the sentence you have to consider the biotic effect on deterioration.

We do agree that the overall deterioration of the materials exposed outside is related to both, biotic and abiotic agents. However, it is usually accepted that “weathering” per-se is associated to changes induced by all but not biological agents. The second is usually defined as “decay “and is associated to the fungi, insects, or bacteria activities. However, following the Reviewer suggestion, additional sentences clarifying the weathering process are provided. Moreover, supplementary information regarding effect of different factors interaction is provided in lines 60-65.

2. Line 50-51. Insert a bibliographic reference that highlights the differences you have explained.

The additional bibliographic reference is inserted.

3. Line 54-55. The sentence is a bit general as sometimes with the ultra violet component of sunlight you can have a loss of yellowing (for example in teak wood see Drevo12004 (woodresearch.sk))

The additional explanation and reference were added providing more details of effect of specific wavelengths on the wood appearance. We would also like to thank for suggestion regarding literature references. Unfortunately, even if intensively searching, we were not able to identify mentioned article in the Wood Research Journal.

4. Line 60. Minimize the use of abbreviations only and add the full name for the abbreviations at least for the first time. CIELAB color space also referred to as L * a * b *

The text was modified as requested.

5. Line 113. Aureobasidium pullulans and A. melanogenum are different species or the same species? Specify if there are multiple species used as biofilms

Aureobasidium melanogenum is the same as Aureobasidium pullulans var. melanogenum (it is the old name of fungus). The biofinish contain one strain of Aureobasidium pullulans that is protected by patents EP1704028 B1 and US7951363 B2

6. Line 139-146. Specify if the starting material have the same origin and cutting age, and how the blocks have been handled (drying, cutting, etc.).

The comment of reviewer is very relevant, however, it is not possible to provide all requested information. Unfortunately, a detailed origin of experimental wood is not known. However, it is assumed that the provenance of Scots pine wood was Scandinavia (Sweden as a main supplied of the raw resources). All the experimental samples were provided by the project partner, following routine industrial procedures for siding boards productions. It included kiln drying of wood (the details regarding the process temperature and duration were not provided). The surface of wooden boards before coating was re-sawed by circular saw. The time duration after sawing and applying the coating was less than two months. As defined in Material and Methods chapter, the coating procedure followed proprietary solution optimized for the siding boards production. Such prepared samples were shipped from the producer by the regular post and were conditioned before weathering in the climatic chamber at 20°C and 65%RH. Same storage conditions were applied after collecting samples along the weathering experiment (every 3 months). Samples were measured with all mentioned methods in a random order to avoid any systematic drift of results.

7. Line 152. Specify how climate data was collected. If dataloggers are used what were the measurement intervals.

The climate data were collected from the local meteorological station, located 500 meters from the weathering stand (San Michele All’Adige, Trentino, Italy).

8. Line 172-173. Provide more information on the equipment used. Is it a portable or benchtop instrument?

Both instruments are portable. Additional information is provided in the revised manuscript.

9. Line180. Attension ® not attention;

The error was corrected, thank you.

10. Line 192. Specify if tangential sections of reduced thickness have been obtained for observation under the microscope;

The microscopic observation was performed directly on the exposed wood surface without additional sample preparation. It was possible thanks to unique configuration of the optical microscope used.

11. Line 216-219. The figure 1 and 2 can be grouped into a single figure;

Thank you very much for the suggestion. However, with your permission we would like to preserve these figures as separate objects. It follows the style of our manuscript, particularly Figure 6 and Figure 7 (also presented separately). Moreover, we do believe that it is better fitting to the journal template as it is more space efficient as well as simplify reading and following presented analysis.

12. Line 390. In fig. 8 (b) is missing;

The missing letter was added.

Thank you very much, again, for all your comments. We do hope that the manuscript has been improved after considering your and other Reviewer suggestions. We do hope that it may be re-considered for publication in the Coatings journal.

With sincerely regards

Jakub Sandak on behalf of authors